# Trophic evolution in ornithopod dinosaurs revealed by dental wear

Attila Ősi [1,2] ✉, Paul M. Barrett [3], András Lajos Nagy[4], Imre Szenti[5], Lívia Vásárhelyi [5], János Magyar[1,2], Martin Segesdi[1,2], Zoltán Csiki-Sava [6], Gábor Botfalvai[1,7] & Viviána Jó[8]

Ornithopod dinosaurs evolved numerous craniodental innovations related to herbivory. Nonetheless, the relationship between occlusion, tooth wear rate, and tooth replacement rate has been neglected. Here, we reconstruct tooth wear rates by measuring tooth replacement rates and tooth wear volumes, and document their dental microwear. We demonstrate that total tooth volume and rates of tooth wear increased steadily during ornithopod evolution, with deeply-nested taxa wearing up to 3360 mm³ of tooth volume/day. Increased wear resulted in asymmetric tooth crown formation with uneven von Ebner line increment width by the Late Jurassic, and in faster tooth replacement rates in multiple lineages by the mid-Cretaceous. Microwear displays a contrasting pattern, with decreasing complexity and pit percentages in deeply-nested and later-occurring taxa. We hypothesize that early ornithopods were browsers and/or frugivores but deeply nested iguanodontians were bulk-feeders, eating tougher, less nutritious plants; these trends correlate with increasing body mass and longer gut passage times.

Dinosaurs were the dominant terrestrial vertebrate herbivores of the Mesozoic Era[1,2]. Among these, ornithopods were the most widespread and abundant herbivores of the Cretaceous Period; these were especially important in Laurasia but also had a significant Gondwanan presence[3,4]. Their success was likely underpinned by sophisticated craniodental adaptations for high-fibre herbivory that matched, or even surpassed, those seen in extant mammalian herbivores. These include the development of novel dental tissues, grinding dentitions, complex jaw movements, and an increase in jaw efficiency, as revealed by studies on jaw musculature, craniodental morphology and biomechanics[5–12], dental microwear[6,13–16], and dental histology[17–20]. These features reached their most advanced development in Late Cretaceous hadrosaurids: consequently, most previous studies have focused on this deeply nested clade. Nevertheless, many of these adaptations first appeared, in incipient form, in earlier-diverging ornithopods, but these animals— non-hadrosaurid iguanodontians (including rhabdodontids and dryosaurids) and *Hypsilophodon*—remain less well-studied. Indeed, several studies have revealed changes in ornithopod dental disparity through time[21], variations in tooth wear and replacement rates[22] and changes in bite efficiency[6,12], all of which show that more basal ornithopod lineages were also experimenting with new ways of mastering herbivorous diet. Consequently, additional research on these non-hadrosaurid taxa will likely provide critical insights into the evolution of hadrosaurid herbivory and contribute substantially to our understanding of Mesozoic palaeoecology and of the varied solutions dinosaurs adapted to the problems posed by eating plants.

¹ELTE Eötvös Loránd University, Institute of Geography and Earth Sciences, Department of Palaeontology, Pázmány Péter sétány 1/C, Budapest 1117, Hungary. ²Hungarian Natural History Museum, Baross u. 13, Budapest 1088, Hungary. ³Fossil Reptiles, Amphibians and Birds Section, Natural History Museum, Cromwell Road, London SW7 5BD, United Kingdom. ⁴Department of Propulsion Technology, Széchenyi István University, Egyetem tér 1, 9026 Győr, Hungary. ⁵University of Szeged, Interdisciplinary Centre of Excellence, Department of Applied and Environmental Chemistry, Rerrich Béla tér 1., 6720 Szeged, Hungary. ⁶Faculty of Geology and Geophysics, University of Bucharest, 1 Nicolae Bălcescu Avenue, 010041 Bucharest, Romania. ⁷HUN-REN–MTM–ELTE Research Group for Paleontology, Pázmány Péter sétány 1/C, Budapest H-1117, Hungary. ⁸ELTE Eötvös Loránd University, Institute of Geography and Earth Sciences, Department of Physical Geography, Pázmány Péter sétány 1/C, Budapest 1117, Hungary. ✉e-mail: osi.attila@ttk.elte.hu

In this work, we present analyses of dental wear traits that reveal previously unknown data on tooth formation time, tooth replacement rates, and daily tooth wear rates for a wide range of taxa spanning the evolutionary history of Ornithopoda. We then combine information on tooth wear rates with observations from 2D and 3D dental microwear to reconstruct changes in ornithopod feeding through time, offering insights into the ecological success of these animals and their potential interactions with contemporary floras.

## Results

### Asymmetrical tooth formation
Significant modifications to tooth crown internal structure first occurred among early-diverging ornithopods. In labiolingual or occlusal thin-sections of non-hadrosaurid ornithopod teeth, the dentine is labiolingually thicker on the side of the crown opposite the cutting edge (=working side) than on the cutting edge side, as measured from the pulp cavity (Fig. 1). This asymmetry is first encountered in the Late Jurassic taxa *Dryosaurus* and *Camptosaurus* (working side 18.6% and 27.9–31.8% thicker, respectively), and is further elaborated in Cretaceous ornithopods (e.g., 37.2% in *Tenontosaurus*, 49.2% in *Mochlodon*; Table 1).

This difference in dentine thickness is not due to an increased number of VELs in iguanodontians, but instead to a greater VEIW on the working side (Table 1), as already pointed out in *Mochlodon*[22]. In the more basal *Hypsilophodon* dentine (and enamel) of the same thickness is present on both sides of the crown (Fig. 1b), and in cf. *Thescelosaurus*, the working side is only 10% thicker than that of the cutting edge.

In taxa with no (or minimal) difference between labial and lingual dentine thickness, individual VEIW is low (10.82–22.77 μm). By contrast, in taxa where there is a marked difference in dentine thickness between the labial and lingual sides (i.e., >25%: e.g., *Tenontosaurus*, rhabdodontids), the VEIW is also much greater (18.21–53.18 μm).

With respect to tooth formation time, the minimum number of VEL (e.g., *Dryosaurus*: 60; rhabdodontids: 80–140, *Iguanodon*: 219; *Edmontosaurus*: 339[17]) increases somewhat with body size and VEIW although this trend cannot be generalised (Table 1).

### Tooth replacement and replacement rate
In non-hadrosaurid ornithopods the functional tooth replacement tooth pairs exhibit a wide range of developmental stages (Fig. 2) forming the *Zahnreihen* defined by Edmund[23]. However, none of the surveyed specimens display a condition wherein a replacement tooth had just become functional (i.e., crown started to wear) and the formation of a new replacement tooth germ had just started. Nonetheless, there are examples of both slightly earlier (replacement tooth 90% erupted and still unworn, with no additional tooth germ under it) and slightly later (tooth just started to function, minimally worn, with a very small replacement tooth under it) eruption stages (Fig. 2). This observation suggests that tooth replacement rate in these ornithopods was largely similar to tooth formation time, the two values differing by (at most) a few days. This is a very important detail, as it allows estimating tooth replacement rates even in cases where CT scans or jaw cross-sections are unavailable, simply by determining the formation time of an isolated tooth.

In contrast to other investigated non-hadrosaurid ornithopods, the Early Cretaceous *Tenontosaurus tilletti* (OMNH 58340)[24] and *T. dossi* (SMU 93132) have some tooth positions that include two replacement teeth below the functional tooth suggesting that these do not represent instances of pathological cases, and also that the formation of two consecutive replacement teeth (i.e., more rapid tooth replacement) was not linked to a specific part of the dentition.

Calculated Z-spacing values in the studied taxa range from 2.11 to 2.43 (Table 1), higher than the values of ~2.0 known for hadrosaurids. We did not find any correlation between Z-spacing and either rate of tooth replacement or wear rate (see below).

### Macrowear pattern and tooth wear rate
The wear facets in our ornithopod sample remain similar in morphology from the Late Jurassic to the Late Cretaceous (e.g., [6,25–28]; Supplementary Fig. 2). In all the taxa studied, wear facet angle with respect to the apicobasal axis (the occlusal plane, *sensu* Ostrom[5]) is conservative, sloping at approximately 40° or more on dentary teeth, while it is between 26 and 40° in hadrosaurids (Supplementary Data 1; A.Ő., pers. obs. 2022). Based on the average wear facet areas calculated from 3D modelling of teeth with different degrees of wear, individual wear facet size is very consistent among similarly sized ornithopods. However, when calculating a total wear facet area for a jaw quadrant, there are significant inter-taxon differences that show strong correlation with changes in body size (Fig. 2b). Compared to small-bodied taxa (e.g., *Dryosaurus*, *Convolosaurus*, *Mochlodon*), tooth number

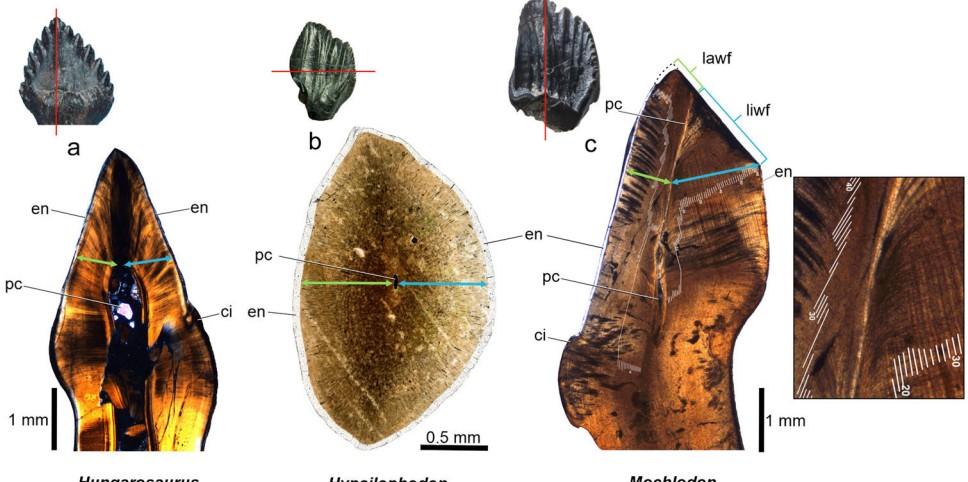

**Fig. 1 | Development of asymmetrical tooth formation in ornithopod dinosaurs.** While the ankylosaur *Hungarosaurus* (**a** labiolingual section) and the non-ornithopod genasaurian *Hypsilophodon* (**b** horizontal section) have tooth crowns with similarly thick VEIBs and enamel on the labial and the lingual sides, in the rhabdodontid *Mochlodon* (**c** labiolingual section) and in other ornithopods the working side of the crown (blue arrow) is always thicker, composed of greater VEIWs. The wear facet is always more extensive on the thicker, working part (blue arrow) of the crown. ci cingulum, en enamel, lawf wear facet labial to the pulp cavity, liwf wear facet lingual to the pulp cavity, pc pulp cavity.

**Table 1 | Dental wear and body mass parameters in taxa used in this study**

| taxon | Higher-level taxon | Age | Working side thickness relative to cutting edge side (%) | Mean VEIW (µm) labial/lingual *estimated **estimated from phylogenetic bracket ***data from Erickson[17] | mean wear facet area (mm²) for one lower jaw quadrant | estimated tooth replacement rate (day) *estimated **estimated from Erickson[17] data from Erickson[17] | Z-spacing | complete tooth crown volume (mm³) *estimated | worn crown volume (mm³) for jaw qudrant/day | Pit% | max. body mass (kg) *estimated |
|---|---|---|---|---|---|---|---|---|---|---|---|
| Dryosaurus altus | Omithopoda | Late Jurassic, Kimmeridgian-Tithonian | 18,6 | 22,77/16,96 | 239,59 | 60 | 2,11 | 288,53 | 46,01 | 71,84 | 169 |
| Camptosaurus dispar | Omithopoda | Late Jurassic, Kimmeridgian-Tithonian | 27,9 – 31,8 | 53,18/? | 532,95 | 75 | 2,33 | 1027,77 | 119,11 | 85,79 | 1323 |
| Dysalotosaurus lettowvorbecki | Omithopoda | Late Jurassic, Kimmeridgian-Tithonian | NA | NA | NA | NA | NA | NA | NA | 89,85 | NA |
| Cumnoria prestwichii | Omithopoda | Late Jurassic, Kimmeridgian | NA | NA | 792,68 | NA | NA | NA | NA | 39,69 | 185 |
| Owenodon hoggii | Omithopoda | Early Cretaceous, Berriasian | NA | NA | 644,67 | NA | 2,21 | NA | NA | 82,08 | 185 |
| Iguanodon sp. | Omithopoda | Early Cretaceous, Barremian | 28,8 | 29,38/? | 4058,12 | 219 | 2,43 | 12472,02 | 861,00 | 57,38 | 4500 |
| Convolosaurus marri | Omithopoda | Early Cretaceous, Aptian | NA | ~15,6** | 275,66 | NA | NA | NA | NA | 60,79 | 150 |
| Tenontosaurus dossi | Omithopoda | Early Cretaceous, late Aptian | NA | NA | 1536,60 | 80 | NA | NA | NA | 76,41 | NA |
| Tenontosaurus tilletti | Omithopoda | Early Cretaceous, Albian | 37,2 | 25,53/34,17 | 1536,60 | 80 | 2,32 | 1605,59 | 146,03 | 70,51 | 1019 |
| Zalmoxes sp. | Omithopoda | Late Cretaceous, Maastrichtian | 29,2-56,8 | 26,33/37,24 | 773,30 | 90 | NA | 388,45 | 15,60 | 61,13 | 185 |
| Mochlodon vorosi | Omithopoda | Late Cretaceous, Santonian | 41,4-49,2 | 18,21/31,11 | 372,00 | 140 | NA | 290,81 | 12,00 | 82,55 | 41 |
| Rhabdodon sp. | Omithopoda | Late Cretaceous, Campanian-Maastrichtian | NA | NA | 1552,21 | 120* | 2,00 | 3377,76 | 171,20 | 73,09 | 947 |
| Matheronodon provincialis | Omithopoda | Late Cretaceous, late Campanian | NA | 24,55/? | 2513,52 | 120 | NA | 9199,07 | 159,50 | NA | 947* |
| Edmontosaurus sp. | Omithopoda | Late Cretaceous, Maastrichtian | NA | 19,8 | 7749,84 | 50** | NA | 4000,00 | 3360,00 | 65,56 | 6610 |
| Maiasaura peeblesorum | Omithopoda | Late Cretaceous, Campanian | NA | NA | NA | 78 | NA | 3800,00 | 2046,15 | 82,32 | 3656 |
| Hypsilophodon foxii | Genasauria | Early Cretaceous, late Barremian | NA | 10,82 | 139,50 | 64 | NA | 61,84 | 8,34 | 58,17 | 24,1 |
| Thescelosaurus sp. | Genasauria | Late Cretaceous, Maastrichtian | 10,6 | 20,37/? | 168,60 | 54 | NA | 35,46 | 5,24 | 76,63 | 108 |
| Edmontonia rugosidens | Ankylosauria | Late Cretaceous, late Campanian | NA | 13,5*** | NA | 279* | NA | 200* | 3,44 | NA | 2256 |
| Hungarosaurus tormai | Ankylosauria | Late Cretaceous, Santonian | NA | 16,97 | 454,44 | 126 | NA | 136,28 | 6,33 | 72,67 | 688 |
| Pinacosaurus grangeri | Ankylosauria | Late Cretaceous, middle Campanian | NA | NA | NA | 73 | NA | 50,00 | 0,27 | NA | 1000 |

**Table 1 (continued) | Dental wear and body mass parameters in taxa used in this study**

| taxon | Higher-level taxon | Age | Working side thickness relative to cutting edge side (%) | Mean VEIW (µm) labial/lingual *estimated **estimated from phylogenetic bracket *** data from Erickson[17] | mean wear facet area (mm²) for one lower jaw quadrant | estimated tooth replacement rate (day) *estimated **estimated *** data from Erickson[17] | Z-spacing | complete tooth crown volume (mm³) *estimated | worn crown volume (mm³) for jaw qudrant/day | Pit% | max. body mass (kg) *estimated |
|---|---|---|---|---|---|---|---|---|---|---|---|
| *Triceratops* sp. | Ceratopsia | Late Cretaceous, Maastrichtian | NA | 15,8*** | NA | 83** | NA | 2650,00 | 791,81 | NA | 12000 |
| *Othnielosaurus* sp. | Ornithischia | Late Jurassic, Kimmeridgian-Tithonian | NA | NA | 97,76 | NA | NA | NA | NA | NA | 16,5 |

Note that in mean VEIWs (µm) labial and lingual data were calculated where tooth crown was asymmetrical and both sides of the dentine were measurable (i.e., VELs and VEIWs could be observed). In the case of teeth where the dentine of the labial and lingual sides was formed symmetrically, only one value was listed.

remains constant in certain larger taxa (e.g., *Tenontosaurus*, *Zalmoxes*) but tooth size gets larger, producing an increased total occlusal surface. In more deeply nested non-hadrosaurids (e.g., *Iguanodon*) both tooth number and absolute tooth size rise, resulting in even larger occlusal surface areas. Conversely, in hadrosaurids, individual tooth size grows only minimally (or not at all) but the number of functional teeth per jaw quadrant increases, resulting in the largest recorded values for occlusal surfaces. During ornithopod evolution wear facet area in a jaw quadrant increased proportionally with body mass (Correlation: 0.85, $p < 0.01$; Fig. 2b).

In taxa for which minimum tooth replacement rate and highest worn tooth crown volume can be calculated, the daily amount of worn tooth crown volume (i.e., daily wear rate) per jaw quadrant is also available (Table 1). For ornithopods, we found a strong (0.95, $p < 0.01$) positive correlation (Fig. 3a) when comparing the $\log_{10}$ value of the daily wear with $\log_{10}$ body mass (body mass data from Benson et al.[29]), demonstrating that ornithopods with a larger body mass wore down a greater daily tooth volume. When data from non-ornithopod ornithischians (ankylosaurs and ceratopsians) are added, the correlation becomes non-significant (0.59, $p = 0.016$). However, occasionally taxa with similar body masses show significantly different wear rates (e.g., daily worn crown volumes in *Dryosaurus*−46 mm³ vs *Zalmoxes* −15.6 mm³, or *Edmontosaurus*−3360 mm³ vs *Iguanodon*−861 mm³); see Fig. 3a). By dividing the daily amount of wear by the average wear facet area per jaw quadrant, we can calculate the approximate tooth crown height worn per day (Table 1); these values are relatively low (20–110 µm) in *Hypsilophodon* and rhabdodontids, but in larger-bodied taxa, such as *Iguanodon* or *Edmontosaurus*, they can reach as much as 212–433 µm. These results are consistent with the wear values reconstructed by Erickson[17] for infant (200 µm/day) and adult (500 µm/day) *Maiasaura* tooth crowns. This implies that while mammal teeth record microwear features generated over several days of feeding (Solounias et al.[30] and Damuth and Janis[31], and references therein), in ornithopods these features reflect a much shorter time span (less than one day, possibly a few hours).

Our analysis also reveals a significant positive correlation (0.93, $p < 0.01$) between total complete tooth crown volume and daily wear rate per jaw quadrant (Fig. 3b), as well as between total complete tooth crown volume and body mass (0.93, $p < 0.01$), reminiscent of the results obtained by Janis[32] for ungulate mammals. These results show that total crown volume increase within Ornithopoda is clearly related to increasing body mass and faster wear rates, indicating a change in diet (see microwear patterns, below).

There is, however, no strong correlation among ornithopods neither for enamel thickness vs total complete tooth crown volume (correlation: 0.63, $p = 0.06$), nor for enamel thickness vs daily wear rate (correlation: 0.58, $p = 0.02$; Supplementary Data 1), and significant differences can occur between similarly sized taxa (e.g., enamel thickness in *Camptosaurus*: 97 µm vs *Tenontosaurus*: 145 µm); even more striking is the difference in this respect between the medium-sized *Matheronodon* (179 µm) and the large-bodied *Edmontosaurus* (155 µm).

### Microwear analysis

For the 16 taxa examined, a total of 326 micrographs were analysed (see Fig. 4a and Supplementary Data 3). The highest pit rates were identified in the Late Jurassic *Dysalotosaurus* and *Camptosaurus* (89.84% and 85.79%, respectively). Pit sizes are very large in *Dryosaurus*, somewhat smaller in *Camptosaurus*, while some of the smallest examples are found in *Dysalotosaurus*. The Late Cretaceous rhabdodontids have quite dissimilar pit-scratch occurrences: while *Mochlodon* shows the highest pit rate among these taxa, along with a few long, well-directed scratches, the wear in *Rhabdodon* teeth is dominated by large pits, and *Zalmoxes* has a much higher number of scratches. By contrast, distribution of microwear features differs only slightly

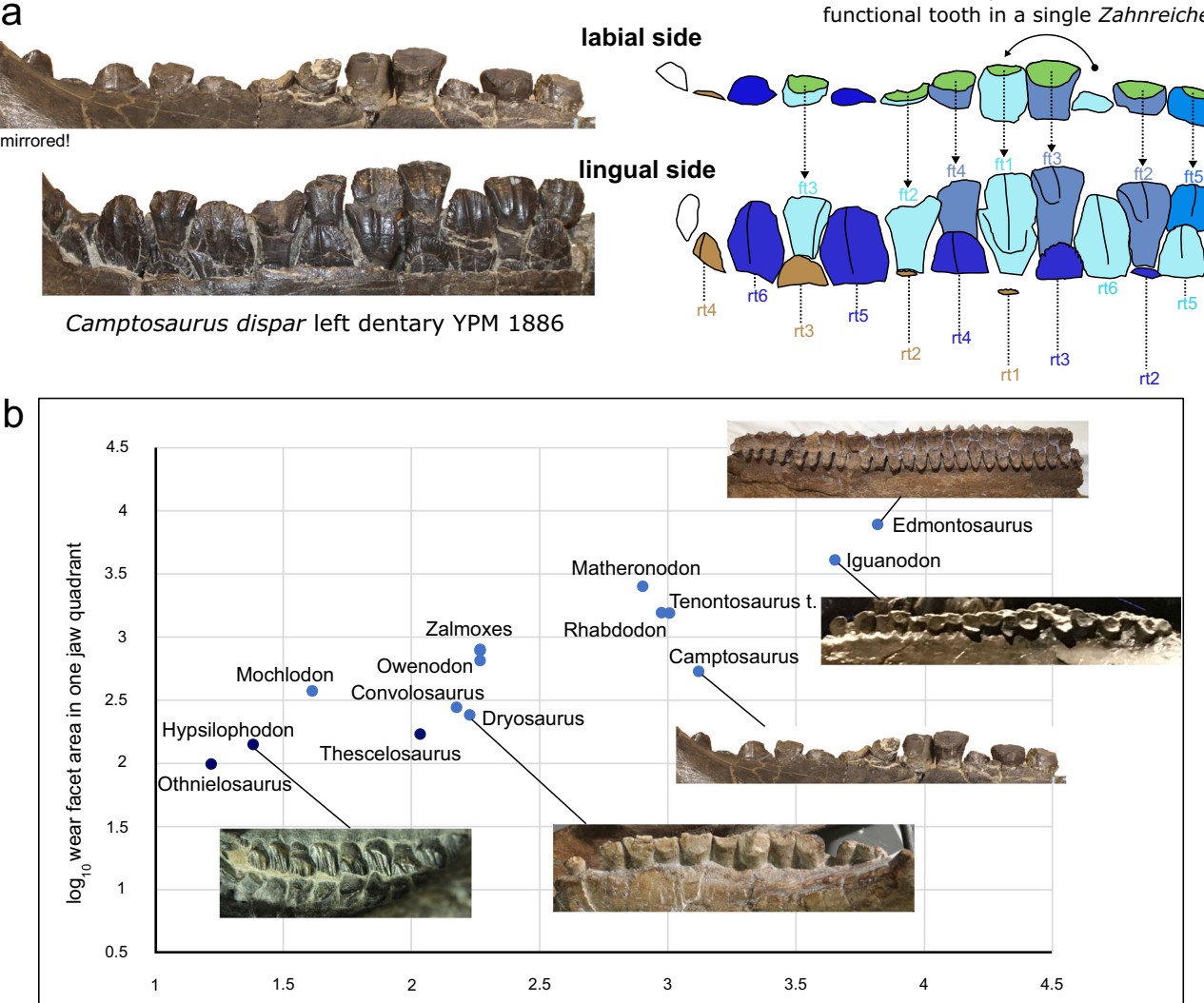

**Fig. 2 | Tooth replacement and change of wear facet area in ornithopods. a** Left lower tooth row of the Late Jurassic *Camptosaurus* (YPM 1886) showing functional (ft1–ft5) and replacement (rt1–rt6) teeth, and the explanatory drawing of the different "Zahnreiche" with the light blue series passing through from replacement to functional teeth. Functional teeth have wear facets (green area). **b** Log$_{10}$ values of the wear facet area vs log$_{10}$ body mass in some non-ornithopod genasaurian (*Othnielosaurus*, *Hypsilophodon*, *Thescelosaurus*, with dark blue) and selected ornithopod taxa (with light blue) (Source data are provided as a Source Data file).

between the two *Tenontosaurus* species, with *T. dossi* showing slightly higher pit rates (76%), and *T. tilletti* (70.5%) having somewhat smaller pits and slightly longer scratches. Our analysis confirms the results of previous tooth wear studies on *Edmontosaurus*[15,16,33], demonstrating a relatively high number of consistently directed, long scratches, with very low numbers of small pits. *Iguanodon* shows slightly fewer scratches and a lower pit rate with larger pits compared to *Edmontosaurus*. PCA plot of 2D microwear features reveals a shift from a higher number of pits in earlier-diverging taxa towards a higher proportion of scratches in more deeply nested ornithopods (Fig. 4a).

The 3D analysis was run on the same set of 326 micrograph areas used in the 2D analysis. PCA shows that *Hypsilophodon* and the Late Jurassic taxa (*Dryosaurus*, *Dysalotosaurus*, and *Camptosaurus*) have the highest complexity (Asfc) values and that juvenile specimens of *Maiasaura* also display high complexity (Fig. 4b and Supplementary Data 3). Meanwhile, complexity values are much lower in other ornithopods (*Owenodon*, *Iguanodon*, *Edmontosaurus*, *Tenontosaurus*). Remarkably, rhabdodontids can also be distinguished from each other based on complexity, with a high value in *Rhabdodon* and very low

values in *Mochlodon* and *Zalmoxes*. Comparing the 2D and 3D data, high 3D complexity of the wear surface correlates well with high pit percentages obtained in the 2D analysis. In terms of anisotropy, the separation between the surveyed taxa is not as significant as it is for complexity. By far, the highest anisotropy value is recorded in *T. tilletti* (in contrast to the lower value found in *T. dossi*), which correlates well with the relatively higher proportion (30%) of similarly oriented scratches identified in this taxon in the 2D analysis.

## Discussion
### Innovations in dentition structure
One of the most striking changes recognised here occurred between early genasaurians and basal ornithopods. Early-diverging ornithischians (Thyreophora, *Lesothosaurus*, *Agilisaurus*, Jeholosauridae, Thescelosauridae) had small, leaf-shaped teeth. Tooth replacement was relatively slow[23], and the volumetric mass of tooth crowns—either in a single alveolus or per jaw quadrant—is relatively small (Table 1). During the Jurassic, more robust, apicobasally elongated, labiolingually broader, rhomboid tooth crowns appeared in ornithopods, the

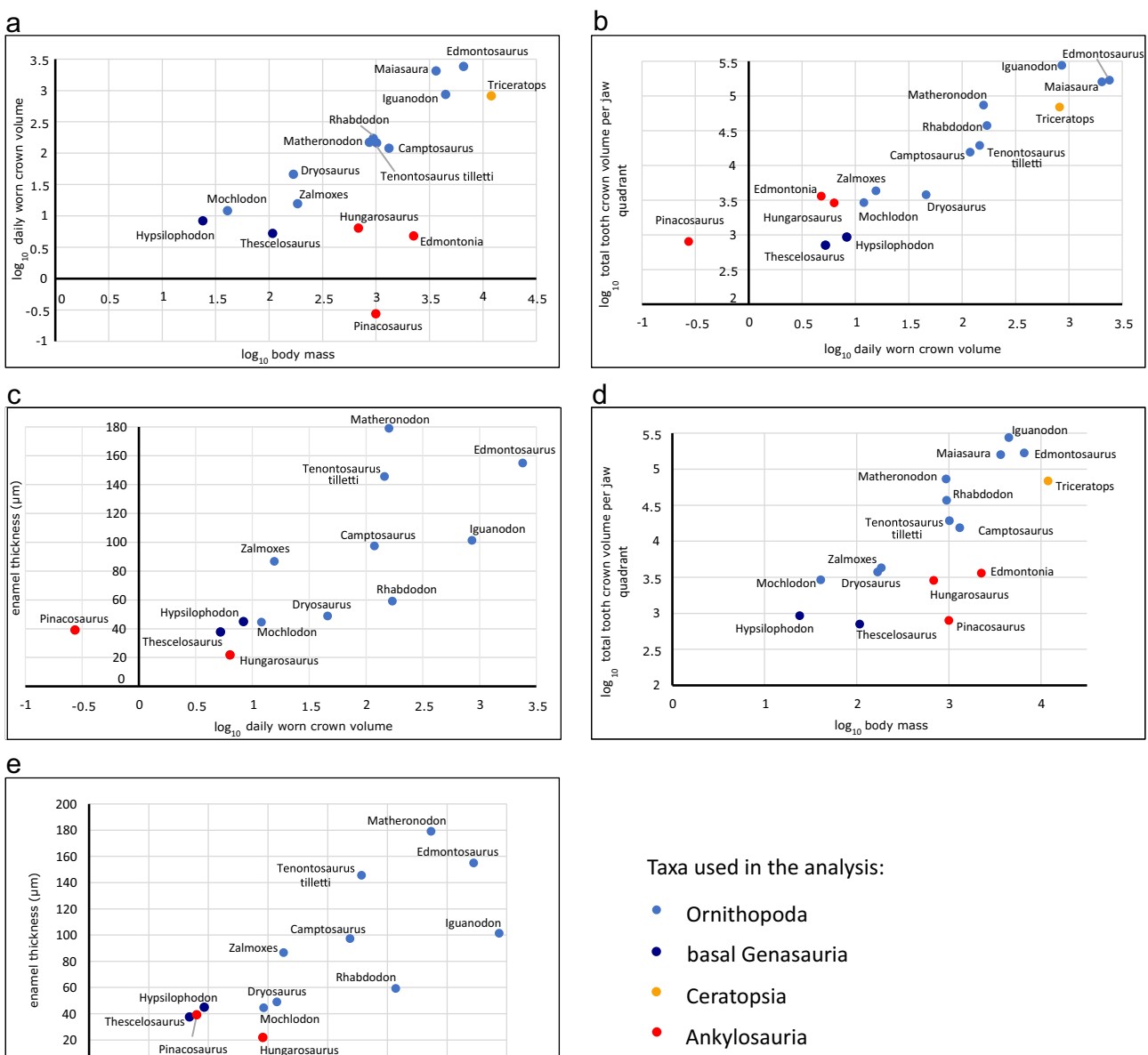

**Fig. 3 | Plots showing the relationship between different dental parameters surveyed in this study among the sampled ornithischian dinosaurs. a** Log10 daily worn crown volume versus log10 body mass. **b** Log10 total tooth crown volume per jaw quadrant versus log10 daily worn crown volume. **c** Log10 daily worn crown volume versus enamel thickness (μm). **d** Log10 total tooth crown volume per jaw quadrant versus log10 body mass. **e** Log10 total tooth crown volume per jaw quadrant versus enamel thickness (μm). (Source data are provided as a Source Data file).

earliest-occurring representatives of which (surveyed in this study) are *Dryosaurus, Dysalotosaurus* and *Camptosaurus*. In these taxa, tooth crown volume relative to alveolar volume becomes much greater, indicating an increase in the amount of worn tooth crown mass (see Fig. 3b). Whereas the total tooth crown volume in a jaw quadrant is 709 mm³ in *Thescelosaurus* (body mass 108 kg), the same parameter is 3750 mm³ in *Dryosaurus* (body mass 169 kg). This difference is even more striking when ornithopods are compared with thyreophorans: the total tooth volume per jaw quadrant is 2861 mm³ in *Hungarosaurus* (body mass 688 kg), but as high as 37,155 mm³ in *Rhabdodon* (body mass 947 kg) (Table 1, Supplementary Data 1).

Late Jurassic ornithopods (*Camptosaurus, Dryosaurus*) show a clear increase in total tooth crown volume relative to body mass (Fig. 3d), a correlation that continues up to the Late Cretaceous. With larger body mass, individual teeth also showed tendency to increase in

relative volume, and this process occurred in several distinct lineages. For example, although tooth number in *Lanzhousaurus*, one of the largest Early Cretaceous iguanodontians, increased only minimally (14 teeth in a jaw quadrant) compared to earlier-diverging iguanodontians (e.g., *Tenontosaurus tilletti*, 12 teeth; rhabdodontids, 8–11 teeth), its tooth crowns are the largest known among ornithischians in terms of both relative width and length[34,35].

Perhaps surprisingly, the functional tooth volume of hadrosaurids was not proportionally greater than in non-hadrosaurid iguanodontians (e.g., *Iguanodon*) (Fig. 3d). However, the significant increase in replacement tooth numbers, the more compact dental batteries that resulted (containing 5–6 tooth generations[18,23]), and the novel dental tissues of hadrosaurids increased the total tooth volume per jaw further, alongside accelerated tooth replacement[5,17,20]. Among ornithopods, hadrosaurids have the highest rates of daily tooth abrasion,

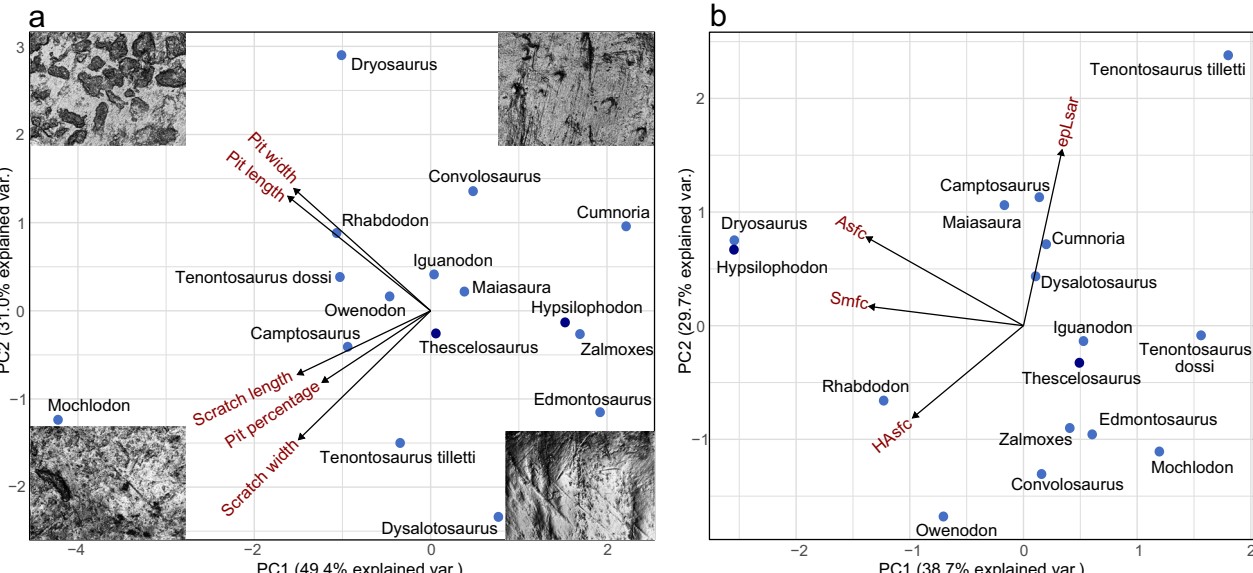

**Fig. 4 | Results of the PCA of 2D and 3D microwear analyses in some non-ornithopod genasaurian (dark blue) and ornithopod dinosaurs (light blue) (Source data are provided as a Source Data file). a** PCA of the scratch and pit parameters. **b** PCA of the four basic 3D parameters. Abbreviations: epLsar, exact- proportion length-scale anisotropy of relief; Asfc, area-scale fractal complexity; Smfc, scale of maximum complexity; HAsfc, heterogeneity of area-scale fractal complexity.

which are 2–4 times greater than in the similarly sized non-hadrosaurid *Iguanodon* (Table 1, Fig. 5). This indicates that *Iguanodon* and hadrosaurids might have consumed different types of plants, with the latter group specialised on more abrasive forage[15,33].

Among rhabdodontids, an increase in tooth crown volume is clearly associated with body mass increase, a trend that peaks in the Campanian-Maastrichtian *Rhabdodon*. *Matheronodon*, however, does not seem to fit this trend. In this species, tooth crowns are extremely widened but, compared to the teeth of *Rhabdodon*, predominantly only mesiodistally. At the same time, the number of maxillary teeth is reduced from 11 (in *Rhabdodon*) to eight (in *Matheronodon*). Although precise body mass data are not available for *Matheronodon*, assuming a similar body weight to that of *Rhabdodon* (947 kg), *Matheronodon* had twice the tooth crown volume per jaw quadrant compared to its relative (73,593 mm³, See Supplementary Data 1). This suggests that the marked tooth crown volume increase observed in this taxon represents the result of some specific feeding-related process that deviated from the trend seen in other rhabdodontids, and complicates the overall trend of positive correlation between body size and tooth crown found in ornithopods. In addition, this distinction further indicates local differences in food preference between closely related sympatric taxa[36], a factor that probably lessened potential interspecific competition between them.

Change in tooth shape is partly caused by the increase of VEIWs in the dentine on the working side of the crown, which disproportionally increases the thickness on that side, leading to marked dental asymmetry. Such a difference was first noted between the teeth of the ankylosaur *Hungarosaurus* and the rhabdodontid *Mochlodon* from the Upper Cretaceous Iharkút site, Hungary[22], but examination of other ornithopods from our sample shows that this asymmetry is also present in the Late Jurassic *Dryosaurus* and *Camptosaurus* as well as in various Early Cretaceous ornithopods. Similar to thyreophorans (such as *Pinacosaurus*[37]), the more basal *Thescelosaurus* and *Hypsilophodon* lack asymmetrical VEIWs, suggesting that development of this structural difference was characteristic of more deeply nested ornithopods with increased tooth crown volume, and that the feature might be a synapomorphy of Ornithopoda.

Among ornithischians, Ceratopsidae[38–40] and Hadrosauridae[5,6,18–20,41] are characterised by more than two tooth generations (1 functional and >1 replacement tooth) in an alveolus at any given time. However, the late Aptian–early Albian *Tenontosaurus tilletti*[24] and the Aptian *T. dossi* each possesses two replacement teeth below the functional tooth in some tooth positions. Since in all recent phylogenetic analyses of Ornithopoda (e.g.,[42–45]), *Tenontosaurus* is only distantly related to, and earlier-diverging than, Hadrosauridae and their close relatives, the limited increase in replacement tooth number seen in the former must represent a convergent acquisition of this feature, independent of its later appearance in hadrosauroids. Furthermore, Hu et al.[46] recently documented two replacement tooth generations in the Barremian-Aptian neornithischian *Jeholosaurus*, indicating that an increase in replacement tooth number was more widespread among neornithischians than previously thought. Remarkably, these two events occurred about the same time as the appearance of the earliest hadrosauroids with two replacement teeth, including the late Aptian/Albian *Altirhinus kurzanovi* from Mongolia[47] and *Probactrosaurus gobiensis* from the Barremian–Albian of northern China and Mongolia[48,49].

Increased numbers of replacement teeth clearly indicate faster tooth replacement, perhaps suggesting a change in the plant food consumed. The most striking floral change taking place in the late Early Cretaceous was the diversification of angiosperms[50–52] and it is tempting to hypothesise that this acceleration of the tooth replacement process, which occurred about the same time concomitantly in several different neornithischian lineages was related to this biotic event; however, such a link remains elusive as correlations between diet, co-occurrence, and ecology are difficult to test[2,21]. Nevertheless, it seems clear that the food consumed by the taxa displaying higher tooth replacement rates must have been more mechanically resistant, at the least, causing teeth to wear more rapidly.

**Correlation between tooth wear rate and microwear pattern**
Comparing tooth wear rates with microwear features across the sampled ornithischian taxa, our study reveals that daily worn crown volume increased steadily in step with increasing body mass during ornithopod evolution (Fig. 5). Conversely, we see decreasing complexity and pit percentages in more deeply nested and/or later-

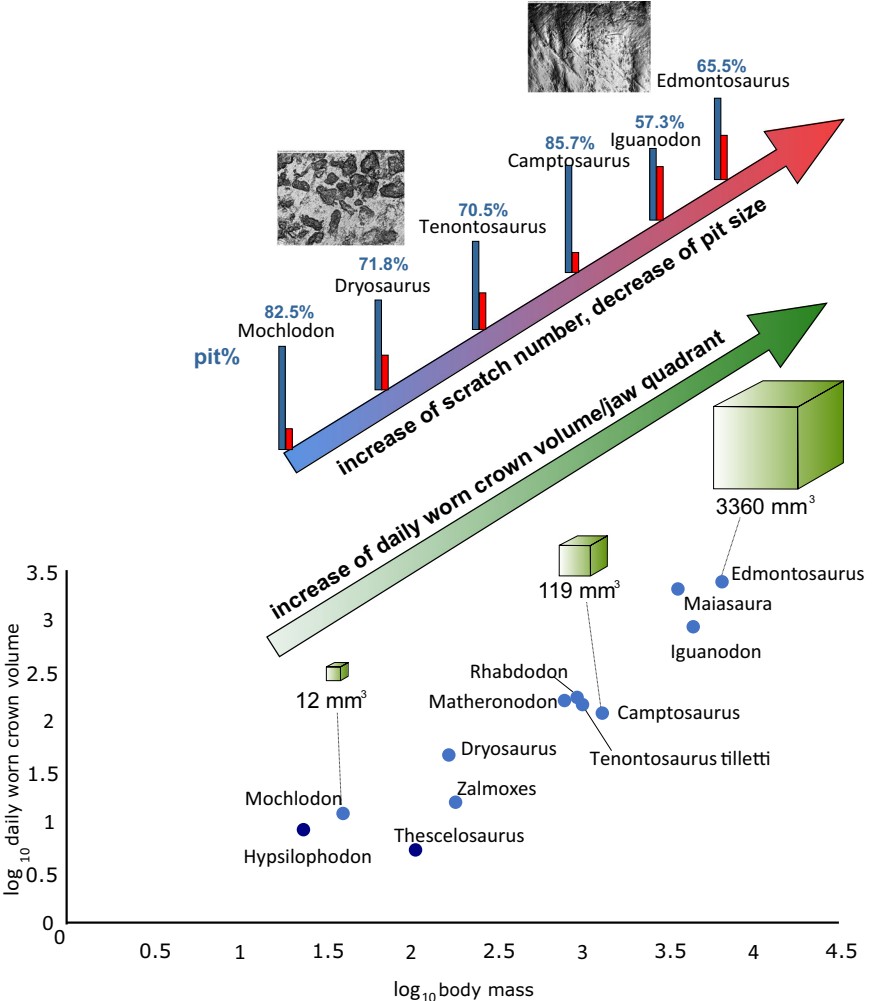

**Fig. 5 | Composite figure illustrating the relationship of daily wear rate vs microwear features in some non-ornithopod genasaurian (dark blue) and selected ornithopod dinosaurs (light blue) (Source data are provided as a Source Data file).** This demonstrates that in correlation with body size increase, the daily amount of worn tooth crown/jaw quadrant also strongly increased during the evolution of ornithopod dinosaurs. In parallel, microwear results reveal a more complex and pit-dominated wear pattern in basal ornithopods vs a less complex and scratch-rich wear pattern found in advanced iguanodontians.

occurring taxa. We infer that the observed changes in wear rates and microwear textures are related. This suggests that during ornithopod evolution, the feeding mechanics and food preference of these animals changed in concert with previously documented increases in body mass[53,54] and changes in tooth morphology[21]. Pits, which are features more typical of browsing and frugivory, dominate microwear in small- to medium-sized early-diverging taxa (*Dysalotosaurus, Dryosaurus, Camptosaurus*), making it likely that these taxa were selective feeders that consumed preferentially the more nutrient-rich plant parts (fleshy leaves, buds, fruits). In more deeply nested, larger taxa (*Tenontosaurus, Iguanodon, Edmontosaurus*) scratches became more dominant (see also refs. 15,16,) and tooth wear rates increased, too. Accordingly, these animals may have relied increasingly on bulk feeding, eating more resistant, less nutritious plant food (most probably including more angiosperms[33]), leading to heightened wear rates (up to 200–500 μm/day) comparable to those found in modern mammalian grazers[17]. This scenario is also consistent with the concurrent increases in body mass in these taxa, which likely led to longer gut passage times that would also have been advantageous for digesting low-quality, fibrous forage. However, this trend may not be general, as in some groups (e.g., rhabdodontids), differences in pit-scratch ratio do not necessarily correlate with body size shifts.

To sum up, compared to non-ornithopod genasaurians, there is a drastic increase in total tooth crown volume and degree of tooth wear in Late Jurassic ornithopods, and this trend escalates further in Cretaceous taxa. However, our data also demonstrate that taxa with largely similar body masses may show marked differences in these values, in terms of both wear facet area (e.g., *Dryosaurus* vs *Zalmoxes*) and volume of worn tooth (e.g., *Iguanodon* vs *Edmontosaurus*), suggesting that the foods these taxa consumed might have differed substantially. The tooth wear rate greatly increased in deeply nested ornithopods, with the amount of wear per jaw quadrant reaching up to 310–500 μm per day in hadrosaurids, indicating that tooth wear patterns record only a few hours of functional use in these dinosaurs.

In addition to ceratopsians and derived hadrosauroids, the more basal ornithopod *Tenontosaurus* also shows the initiation of a trend of increasing number of replacement teeth per tooth position. This step was preceded by the development of an asymmetric tooth crown structure, in which greater VEIWs build up the dentine on the working side of the crown, thus increasing both the occlusal surface and the mechanical resistance of the tooth.

Finally, our data document that during ornithopod evolution, increases in tooth crown volume and dental structural transformations (see also Erickson et al.[18]) were accompanied by correlated changes in

microwear pattern. These correlations suggest a large-scale transitional trend from browsing/frugivory that dominated in early-diverging and often rather small-bodied taxa to bulk feeding on more resistant, less nutritious forage that characterized more deeply nested and usually larger-bodied iguanodontians, documenting a fundamental shift in ornithopod ecology through their evolutionary history.

## Methods

### Field and collection work

Fieldwork providing fossil specimens for this study was carried out only at the sites in Iharkút (Hungary) and Transylvania (Valiora), to which the authors of this article had permission from the institutions responsible for the area. The study of fossil specimens in the collections listed in the Supplementary Information file was carried out after consultation with the listed curators and collection managers and after obtaining permission.

### Material

Specimens used in this study are jaw elements with in situ or associated/isolated teeth, representing two non-ornithopod genasaurians, 13 non-hadrosaurid ornithopods and two hadrosaurids; in addition, one early-diverging ornithischian, three ankylosaurs and one ceratopsian have been used as outgroups in different parts of the analyses (see list of these taxa, with the most relevant variables, in Table 1). Data for the replacement rate and wear rate analyses are in Supplementary Data 1, while calculated volumetric data for the taxa and microwear analyses results are summarised in Supplementary Data 2 and 3, respectively. Here, we follow Kosch and Zanno[36] abbreviations concerning the use of von Ebner incremental lines. Thus, von Ebner lines (VEL) representing the daily cessation of growth[17] are the lines we count to estimate tooth formation time, while the von Ebner line increment width (VEIW) is the spacing between incremental lines (i.e., the amount of dentine deposited). Note that tooth wear rate could not be calculated for all taxa where microwear data were available due to the lack of VEL. The reverse is the case only for *Matheronodon provincialis*, for which VEL information is available based on Godefroit et al.[55], but no microwear analysis was possible.

There is an ongoing debate regarding the inclusion of *Hypsilophodon*, jeholosaurids and thescelosaurids in Ornithopoda; nonetheless, all recent phylogenetic analyses agree that they are either early-diverging ornithopods or lie outside Cerapoda (being equally closely related to ornithopods and marginocephalians) and thus are relevant to our research, regardless of the preferred phylogenetic hypothesis[56–59], as they all possess early examples of features that become established more prominently in deeper-nested ornithopod clades.

### Tooth formation time and tooth replacement rate

Tooth formation time (strictly a minimum value, see below) could only be calculated for those taxa for which dental thin-sections (e.g., *Iguanodon, Hypsilophodon*, rhabdodontids), polished and/or etched crown surfaces (e.g., *Dryosaurus, Camptosaurus, Tenontosaurus*), or well-preserved occlusal surfaces formed by dental wear (e.g., *Iguanodon*) were available. This allowed the counting of VEL formed daily during tooth growth within the dentine[17] (see Supplementary Fig. 1); these counts, in turn, offer an estimate of tooth formation time (Table 1). However, it should be noted that these estimates are minimum values, as some VELs are not captured in all thin-sections[59].

Due to preservational biases and collection management protocols it was not possible to use the invasive method of Erickson[17] to count VELs in the functional and replacement teeth of a single tooth family. Moreover, the non-invasive method of D'Emic et al.[60] could not be used in these ornithopods due to a lack of either CT scans or naked-eye observations on functional+replacement tooth pairs. For those

ornithopods where data on tooth replacement rates were unavailable, we used the minimum tooth formation time calculated from VELs as a suitable proxy for tooth replacement rate, since in these taxa there was, almost invariably, a single replacement tooth that started to form at the moment the previous tooth became functional (see Supplementary Fig. 1, below). In all our observations, we used either already worn, functional teeth or the largest intact tooth available for the taxon.

Jaw elements of *Mochlodon* and *Zalmoxes* were scanned using a Bruker Skyscan 2211 micro-CT scanner (Skyscan, Bruker, Belgium) with X-ray source settings of 150 and 180 kV source voltage, 100 μA source current, and 40 ms exposure time. The samples were measured in high power mode using a 3 Mp active pixels CMOS flat panel X-ray detector at 30 and 56 μm voxel resolution.

Z-spacing, as a possible factor for indicating tooth replacement rate[61], has been calculated in most studied taxa. As CT scans are not available for most of the ornithopods studied, with only some parts of the replacement teeth visible, replacement stages were not determined according to Fastnacht's[61] scheme, but instead according to predefined stages, as done in a study of basal ceratopsians[62].

### Calculation of wear facet area, and crown volume

To calculate the wear surface area (mm$^2$) and tooth crown volume (mm$^3$), samples were scanned with a Polyga HDI Compact S1 3D scanner. Before scanning, the specimens were coated with talcum powder to reduce reflection from their surfaces. Specimens were scanned on a turntable connected to the scanner. The point clouds from the multidirectional scans (from at least eight directions −45°) were merged and surfaces created in FlexScan3D v. 3.3.21.8. The resulting 3D polygon files had polygon lengths of 30–80 μm. Further operations, including measurements of worn surface area and crown volume, were performed in Geomagic Wrap v. 2017.0.2.18 (3D Systems, Rock Hill, SC). Total tooth crown volume was obtained by measuring intact, unabraded teeth. Abraded tooth volume was calculated based on the largest and most abraded tooth available for a given taxon by subtracting the volume of an abraded tooth from the volume of a similarly sized intact tooth following Ősi et al.[22]. This value was multiplied by the number of alveoli per jaw quadrant to estimate the amount of tooth material lost through wear per jaw quadrant in one tooth replacement cycle. Dividing the amount of tooth wear by the tooth replacement rate, we obtained the amount of tooth material worn per jaw quadrant per day. This value divided by the wear area gives the average daily amount (thickness in microns) of tooth crown material worn off.

### Microwear analysis

Both traditional (i.e., 2D, feature-based) and dental microwear texture (DMT: 3D) analyses were conducted. Following the procedures outlined in Ősi et al.[22], micrographs were taken only from the dentine close to the enamel-dentine boundary in all measured specimens, as the enamel is too thin (enamel thickness is thinner than the micrograph size in many cases) and poorly preserved to retain features of interest. Sixteen genasaurian taxa (two non-ornithopod genasaurians, 12 non-hadrosaurid and two hadrosaurid ornithopods) were analysed (Table 1, Supplementary Data 3), with an average of 9.5 teeth surveyed per taxon. For both traditional and DMT analyses, we usually sampled three micrograph areas per tooth (with some variation between teeth) and the same areas were analysed by both techniques (Supplementary Data 3). In specimens for which direct micrographs could not be taken, we made high-resolution moulds from the teeth, following procedures described by Grine[63] for hominids. Teeth were first cleaned with cotton swabs soaked with ethyl alcohol, then moulds were made using Coltene President Jet Regular (polysiloxane vinyl) impression material. These moulds were used for making EPO-TEK 301 or Araldite 2020 epoxy resin casts. This technique allows the reproduction of features

with a resolution of a fraction of a micron[64]. In traditional microwear analysis, features are categorised as either scratches or pits. Following Ungar[65], pits are defined as having length-width ratios <4:1, whereas for scratches this ratio is >4:1. Micrographs were taken with a Leica DCM3D confocal microscope (Széchenyi István University, Győr, Hungary). For 2D microwear analysis we used micrographs as grey-scale images with intensity data from the confocal dataset and a resolution of 768 × 576 pixels, corresponding to a 637 × 477 µm field of view. Measurements were carried out using a Leica HC PL Fluotar EPI 20X lens. Images of the micrographs were analysed using Microware v. 4.0, following the procedure described by Ungar[65]. The 768 × 576 pixel grayscale images generated were analysed on a 27" Full HD monitor corresponding to a physical image size of ~24 × 18 cm (assuming a pixel density of 81 pixels per inch) when viewed at 1:1 scale. The micrographs were scaled 1:1 in Microware before counting the identified features. 2D microscopy analysis was performed by two operators in parallel. Initially, several test-micrographs were analysed by both operators (following the protocol of Szabó and Virág[66]), which identified differences between their counting within 5%. Four variables were documented: (1) the percentage incidence of pitting; (2) mean scratch width; (3) mean pit width; and (4) mean pit length. We also documented the number of features measured and the standard deviation of means (Supplementary Data 3).

For the DMT analysis, 3D topographic data from the wear surfaces were collected using the same confocal technique. Raw measurements were post-processed in Mountains v. 8. Datasets were levelled using a least-squares plane method levelling algorithm. Non-measured points were filled with a smooth spline method. No additional data processing was done before surface analysis. This approach was chosen to accelerate 3D analysis and to avoid potential misinterpretation of surface features, minimise subjectivity, and increase reproductivity. A 500 × 500 pixel area was extracted from each micrograph for evaluation purposes. Each 3D topographic dataset was analysed by scale-sensitive fractal analysis (SSFA) based on several previous studies (e.g., refs. [67–71]). In the present study, SSFA attributes measured were anisotropy (epLsar=exact-proportion length-scale anisotropy of relief), complexity (Asfc=area-scale fractal complexity), scale of maximum complexity (Smc), and heterogeneity of area-scale fractal complexity (HAsfc(9 × 9)). In addition, ISO 25178 functional (Sxp), volume (Vvv) and feature (Sha, Shv, Sda, Sdv) parameters were also applied, as a previous study on ornithopods[41] has shown that these can be relevant in DMT analyses. 3D microwear parameters are summarised in Supplementary Data 3.

For visualisation purposes, non-measured points were filled using a smooth shape calculated from the neighbouring points. The resulting surfaces were exported in '.sur' format. A MATLAB algorithm was used to create an automated export of 3D pseudocolor topography maps of the micrographs. For a multivariate analysis of the 2D and DMT data (Supplementary Data 3), a principal components analysis (PCA) was conducted using the *prcomp()* (within 'stats' package, 'scale' and 'center' arguments = TRUE), and *summary()* functions in R (R v. 4.0.5[41]). PCA plots were made with *gbiplot()* function ('ggbiplot' package).

### Reporting summary
Further information on research design is available in the Nature Portfolio Reporting Summary linked to this article.

## Data availability
The microwear and 3D data[72] generated in this study have been deposited in the Zenodo database under accession code https://doi.org/10.5281/zenodo.11092497. The dental wear and tooth replacement data generated in this study are provided in the Supplementary Information/Source Data file. Source Data for Figs. 2–5 can be found in Source Data file. Collection data on specimens are listed in the Supplementary Information file. Source data are provided with this paper.

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

## Acknowledgements

We are grateful to the 2000-2022 field crew for their assistance in the Iharkút fieldwork. Laboratory work was supported by the MTA-ELTE Lendület Dinosaur Research Group (Grant no. 95102 to A.Ő.), Hungarian Scientific Research Fund and National Research, Development, and Innovation Office (NKFIH K 116665 and K 131597 to A.Ő.; FK 146097 to G.B.), National Geographic Society (Grant No. 7228-02, 7508-03 to A.Ő.), Hungarian Natural History Museum, Eötvös Loránd University, the Jurassic Foundation, HUN-REN Hungarian Research Network, and the Hungarian Dinosaur Foundation. This is HUN-REN-MTM-ELTE Paleo contribution No. 405. We are grateful to the following colleagues for helping us in taking samples and 3D scanning: Susannah Maidment (Natural History Museum, London, UK), Eric Buffetaut (CNRS, Paris, France), Hannah Keller and Daniel Brinkman (Peabody Museum, Yale University, New Haven, USA), Matthew Lamanna and Amy Henrici (Carnegie Museum of Natural History, Pittsburgh, USA), Matthew Carrano and Hans-Dieter Sues (Smithsonian National Museum of Natural History, Washington, USA), Louis Jacobs and Dale Winkler (Shuler Museum of Paleontology, Dallas, USA), Annelise Folie and Pascal Godefroit (Muséum des Sciences naturelles de Belgique, Bruxelles, Belgium), Fidel Torcida Fernández-Baldor (Museo de Dinosaurios, Salas de los Infantes, Spain), Didier Clavel (Musée de Cruzy, Cruzy, France), Patrick and Annie Mechin (Vitrolles, France), Oliver Rauhut (Bayerische Staatssammlung für Paläontologie und Geologie, Munich, Germany), Felix Augustin and Henrik Stöhr (Universität Tübingen, Tübingen, Germany), Hilary Ketchum (Oxford University, Museum of Natural History), Xavier Valentin (Musée du Moulin Seigneurial, Velaux, France), Alana Gishlick and Meng Jin (American Museum of Natural History, New York, USA). We are grateful to Peter Ungar and Diane Serenson-Ungar who helped us a lot during our research trip in the USA.

## Author contributions

A.Ő. and P.B. conceived the study, A.Ő., V.J., J.M., M.S. collected measurements and numerical data, G.B., J.M. collected the Transylvanian fossils, A.Ő., M.S., V.J., A.L.N., I.S.Z., L.V. performed analyses, A.Ő., P.B., Z.C.S.S. interpreted results and wrote the first version of the manuscript, and all authors critically edited the manuscript.

## Funding

## Competing interests

The authors declare no competing interests.
