## [Peer Review File · Nature Communications]

Trophic evolution in ornithopod dinosaurs revealed by dental wearReviewers' Comments:

Reviewer #1:

Remarks to the Author:

Dear Editor, Dear Authors,

Thank you for the opportunity to review the manuscript by Attila Ósi and colleagues. The manuscript is well written and well organized, and addresses an interesting research topic. It is a nice contribution based on quite extensive first hand observations that will undoubtedly serve as a valuable source of data for future studies.

The study is actually very straightforward and I have very little to add to the text itself. I would perhaps just reword some things:

p. 4, lines 89, 90: While I agree that Jeholosaurids and Thescelosaurids (whatever these groups comprise!) are relevant to the study, I wouldn't consider them to lie on the ornithopod 'stem'. There is no such thing, even if we write it in quotation marks. Jeholosaurus and Thescelosaurus (and their close relatives) are either early-diverging ornithopods or lie outside Cerapoda (being equally closely related to ornithopods and marginocephalians). I would perhaps write that their teeth may be close to the ancestral ornithopod condition.

Similarly, Hipsilophodon is described in the paper as a "non-ornithopod genasaurian" while, in fact, we don't know whether it is an early-diverging ornithopod or a non-cerapodan neornithischian.

p. 15, line 350: Perhaps replace "ancestors" with a more appropriate term; (1) they are obviously not ancestors and (2) it doesn't look appropriate to speak of an Early Cretaceous taxon (Lanzhousaurus) and its Late Cretaceous "ancestors" (rhabdodontids). :) You speak of "earlier-diverging iguanodontians".

Needless to say, these comments are just nitpicking. If there was something I don't like about the paper, it would be the figures. This study has a great potential to provide a very nice graphical story of the ornithopod trophic evolution. The submitted figures would be appropriate for a supplement.

Finally, I cannot access the data submitted to the Zenodo repository but I assume that the files are the same as those that were included in the supplements.

I recommend acceptance after some very minor modifications to the text and look very much forward to seeing the study published.

Sincerely,
Daniel Madzia

Reviewer #2:

Remarks to the Author:

This is an interesting paper examining evolutionary trends in ornithopod dinosaurs using characteristics of the teeth, specifically tooth wear. It will be of significance to other paleobiologists studying dinosaur evolution, and more broadly to other researchers who may wish to use these methods to examine evolutionary trends in other groups of toothed vertebrates. The data appear to be sound, however, the link to the data repository is not active so I could not check if images of all microwear analyses are available with the paper. If that dataset is not available, I would suggest including a supplementary figure showing where all enamel samples were taken from (for example, as in Kubo et al. 2023 Fig 1). Is it actually just enamel being sampled? It can sometimes be confusing due to wear, and dentine might provide a different signal. It would also be good to know the developmental stage of each tooth being measured. I assume they are quite mature if they have wear on them, but this should be clarified.

I have a few other comments:

- Line 37: I don't 'unevenly thickened von Ebner bands' is correct here. It is the spacing between incremental lines (the amount of dentine deposited) that is thicker. The incremental line of von Ebner is just the daily cessation of growth. In light of this, the abbreviation used (VEIB) is incorrect (it is not a band). You could use von Ebner Line Increment Width, VEIW, following Kosch and Zanno 2020 or Mean Increment Width following Erickson 1996 (dinosaur paper).

-Figure 2b caption: indicate 'dark blue' and 'light blue' for each dinosaur group in the plot.

-Figure 4 caption: what are the abbreviations?

-All figures: genera should be italicized

-The supplementary figures need figure captions. I am not sure why there are drawings of Iguanodon in this figure. Shouldn't this figure have images of the actual specimens examined, or is this available in the online repository?

Reviewer #3:

Remarks to the Author:

Ósi et al publish a splendid, well-written work on the evolution of feeding strategies based on dental features (occlusion, tooth wear rate, tooth replacement rate) in ornithopod dinosaurs. The authors have provided an extensive dataset, comprising the species for which such data could be acquired, and I congratulate them on the methodology used.

The results are of particular interest: yes, the main result being that, during their evolution, ornithopod feeding mechanics and food preference changed together with increase of body mass and variation of tooth morphology was somewhat already thought in the past, but now the authors have provided a quantification to support such hypothesis. The anomalous case of Matheronodon in the analysis further open new windows for research and fieldwork to acquire more data on

rhabdodontid evolution and diversity.

I wasn't aware of the timing of writing and submission of this paper, so I was ready to suggest the authors to include -for a minor revision- the work on Jeholosaurus tooth replacement pattern published by Hu et al. just very recently, but I was extremely glad to see this paper been already mentioned in the manuscript. Indeed, the discovery of a second generation of replacement teeth in such a basal ornithopod opens new insight onto the evolution of this trait -as reported in this manuscript, together with Tenontosaurus-.

The results are well supported by the methodology, and I have nothing to say over the final considerations of the data. The methods are well explained, and I'm looking forward to see such guidelines used for other dinosaurs, especially for new ornithopods in the future.

In conclusion, no revisions are needed, in my opinion.

Reviewer #1

Thank you for the opportunity to review the manuscript by Attila Ósi and colleagues. The manuscript is well written and well organized, and addresses an interesting research topic. It is a nice contribution based on quite extensive first hand observations that will undoubtedly serve as a valuable source of data for future studies. The study is actually very straightforward and I have very little to add to the text itself. I would perhaps just reword some things:

p. 4, lines 89, 90: While I agree that Jeholosaurids and Thescelosaurids (whatever these groups comprise!) are relevant to the study, I wouldn't consider them to lie on the ornithopod 'stem'. There is no such thing, even if we write it in quotation marks. Jeholosaurus and Thescelosaurus (and their close relatives) are either early-diverging ornithopods or lie outside Cerapoda (being equally closely related to ornithopods and marginocephalians). I would perhaps write that their teeth may be close to the ancestral ornithopod condition. Similarly, Hypsilophodon is described in the paper as a "non-ornithopod genasaurian" while, in fact, we don't know whether it is an early-diverging ornithopod or a non-cerapodan neornithischian.

Corrected according to the reviewers comment, we have added a comment noting the labile phylogenetic positions of these taxa and justify why their inclusion helps to establish the primitive condition for the more derived ornithopod taxa that form the basis for most of our analytical results.

p. 15, line 350: Perhaps replace "ancestors" with a more appropriate term; (1) they are obviously not ancestors and (2) it doesn't look appropriate to speak of an Early Cretaceous taxon (Lanzhousaurus) and its Late Cretaceous "ancestors" (rhabdodontids). :) You speak of "earlier-diverging iguanodontians".

Corrected according to the reviewers comment, we have used the terms deeply-nested and early-branching (and other similar equivalents) where necessary throughout and removed the word ancestors.

Needless to say, these comments are just nitpicking. If there was something I don't like about the paper, it would be the figures. This study has a great potential to provide a very nice graphical story of the ornithopod trophic evolution. The submitted figures would be appropriate for a supplement.

Thankyou for the comment. While we would have liked to add more illustrations, the figures show particularly important aspects of the research itself and we think they are the most informative.

Finally, I cannot access the data submitted to the Zenodo repository but I assume that the files are the same as those that were included in the supplements.

We can only make the Zenodo link active after the MS will be accepted, unless anyone can download our raw data files before the publication. To solve this problem we created a google drive link with all the supplementary files (3D models of the teeth used and 2D and 3D microwear images).

Reviewer #2

The data appear to be sound, however, the link to the data repository is not active so I could not check if images of all microwear analyses are available with the paper. If that dataset is not available, I would suggest including a supplementary figure showing where all enamel samples were taken from (for example, as in Kubo et al. 2023 Fig 1).

We can only make the Zenodo link active after the MS will be accepted, unless anyone can download our raw data files before the publication. To solve this problem we created a google drive link with all the supplementary files (3D models of the teeth used and 2D and 3D microwear images).

Is it actually just enamel being sampled? It can sometimes be confusing due to wear, and dentine might provide a different signal.

We have clarified this in the manuscript. We've taken the micrographs in all taxa and samples from the hardest mantle dentine, i.e. the closest to the enamel-dentine boundary, since the enamel is too thin in many samples (enamel thickness on the wear facet is simply thinner than the micrograph size) and poorly preserved (fragmented at the cutting edge) to retain features of interest.

It would also be good to know the developmental stage of each tooth being measured. I assume they are quite mature if they have wear on them, but this should be clarified.

We've added a short note in the MS to clarify this question. Concerning the developmental stage of the teeth, we've sampled the largest available specimens on record for each taxon. We used either already worn, functional teeth or the largest one from among the intact teeth available. Worn functional teeth were completely developed at that moment when used. These teeth were used for both the volumetric measurements and for the dental wear analysis. In Supplementary data 2, summarizing the volumetric measurements we documented which of the teeth were worn or complete.

- Line 37: I don't 'unevenly thickened von Ebner bands' is correct here. It is the spacing between incremental lines (the amount of dentine deposited) that is thicker. The incremental line of von Ebner is just the daily cessation of growth. In light of this, the abbreviation used (VEIB) is incorrect (it is not a band). You could use von Ebner Line Increment Width, VEIW, following Kosch and Zanno 2020 or Mean Increment Width following Erickson 1996 (dinosaur paper).

We corrected this in the MS and follow Kosch and Zanno's (2020) abbreviations concerning the using of von Ebner incremental lines (VEL vs VEIW). Accordingly, we have modified all relevant abbreviations in the text and tables.

-Figure 2b caption: indicate 'dark blue' and 'light blue' for each dinosaur group in the plot.

Corrected.

-Figure 4 caption: what are the abbreviations?

We've added explanations for the abbreviations to the figure caption.

-All figures: genera should be italicized.

We have not italicised the genus names on the diagrams because we thought they would be easier to read in regular font at this size.

-The supplementary figures need figure captions.

We've added figure captions to the supplementary figures.

*- I am not sure why there are drawings of *Iguanodon* in this figure. Shouldn't this figure have images of the actual specimens examined, or is this available in the online repository?*

In Supplementary figure 2, for *Iguanodon* spp. we use illustrations from Norman (1980, 1986) because we did not have good photographs of *M. atherfieldensis* and the original specimen illustrated by Norman is fragmentary. In the case of *I. bernissartensis*, although we have our own photographs, the dentition is not clearly visible on these because of the presence of a thick layer of conservation material and the fragmented teeth. Overall, the teeth, especially the replacement teeth, are more comparable in the drawings.

Reviewer #3

Ósi et al publish a splendid, well-written work on the evolution of feeding strategies based on dental features (occlusion, tooth wear rate, tooth replacement rate) in ornithopod dinosaurs. The authors have provided an extensive dataset, comprising the species for which such data could be acquired, and I congratulate them on the methodology used.

*The results are of particular interest: yes, the main result being that, during their evolution, ornithopod feeding mechanics and food preference changed together with increase of body mass and variation of tooth morphology was somewhat already thought in the past, but now the authors have provided a quantification to support such hypothesis. The anomalous case of *Matheronodon* in the analysis further open new windows for research and fieldwork to acquire more data on rhabdodontid evolution and diversity.*

*I wasn't aware of the timing of writing and submission of this paper, so I was ready to suggest the authors to include -for a minor revision- the work on *Jeholosaurus* tooth replacement pattern published by Hu et al. just very recently, but I was extremely glad to see this paper been already mentioned in the manuscript. Indeed, the discovery of a second generation of replacement teeth in such a basal ornithopod opens new insight onto the evolution of this trait -as reported in this manuscript, together with *Tenontosaurus*-.*

The results are well supported by the methodology, and I have nothing to say over the final considerations of the data. The methods are well explained, and I'm looking forward to see such guidelines used for other dinosaurs, especially for new ornithopods in the future.

In conclusion, no revisions are needed, in my opinion.

Thank you for these exceptionally positive comments, we are glad you enjoyed reading the paper!

Reviewers' Comments:

Reviewer #2:

Remarks to the Author:

The authors have addressed all reviewer comments and the manuscript is in good shape. The methods are more clear and the terminology is more accurate.